# PVA-Based Electrospun Materials—A Promising Route to Designing Nanofiber Mats with Desired Morphological Shape—A Review

**DOI:** 10.3390/ijms25031668

**Published:** 2024-01-30

**Authors:** Gizem Ceylan Türkoğlu, Niloufar Khomarloo, Elham Mohsenzadeh, Dilyana Nikolaeva Gospodinova, Margarita Neznakomova, Fabien Salaün

**Affiliations:** 1Department of Textile Engineering, Dokuz Eylul University, İzmir 35397, Turkey; gizem.turkoglu@deu.edu.tr; 2Univ. Lille, ENSAIT, ULR 2461-GEMTEX-Génie et Matériaux Textiles, F-59000 Lille, France; niloufar.khomarloo@junia.com (N.K.); elham.mohsenzadeh@junia.com (E.M.); 3Univ. Lille, ENSAIT, ULR 2461-GEMTEX-Génie et Matériaux Textiles, Junia, F-59000 Lille, France; 4Faculty of Electrical Engineering, Department of Electrical Apparatus, Technical University of Sofia, 1156 Sofia, Bulgaria; dilianang@abv.bg; 5Faculty of Industrial Technology, Department of Material Science and Technology of Materials, Technical University of Sofia, 1000 Sofia, Bulgaria; mpn@tu-sofia.bg

**Keywords:** poly(vinyl alcohol), electrospinning, nanofiber, fiber morphology, electrospun mats, biomedical applications

## Abstract

Poly(vinyl alcohol) is one of the most attractive polymers with a wide range of uses because of its water solubility, biocompatibility, low toxicity, good mechanical properties, and relatively low cost. This review article focuses on recent advances in poly(vinyl alcohol) electrospinning and summarizes parameters of the process (voltage, distance, flow rate, and collector), solution (molecular weight and concentration), and ambient (humidity and temperature) in order to comprehend the influence on the structural, mechanical, and chemical properties of poly(vinyl alcohol)-based electrospun matrices. The importance of poly(vinyl alcohol) electrospinning in biomedical applications is emphasized by exploring a literature review on biomedical applications including wound dressings, drug delivery, tissue engineering, and biosensors. The study also highlights a new promising area of particles formation through the electrospraying of poly(vinyl alcohol). The limitations and advantages of working with different poly(vinyl alcohol) matrices are reviewed, and some recommendations for the future are made to advance this field of study.

## 1. Introduction

Poly(vinyl alcohol) (PVA) is a synthetic polymer bearing CH_2_CH(OH) repeating units. This polymer composed of vinyl monomers is not prepared directly from vinyl alcohol because of its instability, which induces a tautomerization mechanism to acetaldehyde. It is obtained following a post-polymerization reaction of a homopolymer composed of protected monomer units such as vinyl acetate, vinyl ester, or vinyl ether. The first patent for the preparation of PVA dates back to 1924 by W.O. Herrmann et al., where a solution of PVA was obtained by saponifying poly(vinyl ester) with a caustic soda solution [1]. Since then, the production of this polymer has grown steadily to become a plastic material used in many fields of application, with an annual growth rate of over 4% for 2030. PVA can be obtained through various synthetic routes, of which the method developed by Hermann and Haehnel remains the most used. This process is based on the radical polymerization of vinyl acetate followed by the hydrolysis of the acetate groups in the presence of a strong base in methanol [2]. The resulting product, for which its physico-chemical properties depend on the degree of polymerization and the hydrolysis kinetics, is then precipitated, washed, and dried. Vinyl ester monomers can also be used as PVA precursors in the same way as vinyl acetate [3]. Using vinyl ethers (CH_2_=CHOR) via cationic polymerization using a Lewis acid catalyst results in a homopolymer hydrolyzed under acidic conditions to form PVA [4]. This process preferably leads to the obtainment of an isotactic PVA. Another possibility described in the literature is to use an aldol group transfer polymerization from silylated vinyl aldehydes and ethers (CH_2_=CHOSiR_3_) to produce a silylated vinyl alcohol polymer with a terminal aldehyde group [5].

PVA is a synthetic and semicrystalline polymer with multi-hydroxyl groups, with an excellent oxygen barrier, dyeing properties, and mechanical strength [6,7,8]. The physicochemical properties of PVA are closely related to its preparation method and, in particular, to the state of hydrolysis, which can be either complete or partial. The synthesis conditions allow for different molar masses, solubility, and adhesion or mechanical characteristics. Conventional two-figure PVA grade nomenclature includes information on the parameters affecting the properties of this polymer in solution, namely the apparent viscosity value of the concentrated solution at 4 wt.% at 20 °C, and the degree of hydrolysis (DH) of the polymer. Thus, PVA-10-98 means that this grade has a viscosity of 10 mPa.s, and is 98% hydrolyzed [9]. These two parameters are known to have a significant influence on the processability and performance of this polymer. Thus, as the degree of hydrolysis increases, so do the crystallinity, melting temperature, and mechanical strength due to the high level of hydrogen bonding between the chains [10,11,12]. A lower grade of hydrolysis leads to higher solubility in water and may have better compatibility with other excipients. Its unique properties make it the material of choice for various industries and application areas. It has good mechanical performance, high tensile strength and flexibility, and oxygen and aroma barrier properties. In addition, it is odorless and non-toxic and has excellent film-forming, emulsifying, and adhesive properties. It also has good tolerance to grease, oil, and a wide range of solvents. However, as a water-soluble polymer, these properties depend on moisture levels. Thus, water molecules act as a plasticizer, decreasing tensile strength while increasing its elongation at break. It is a semi-crystalline and biodegradable polyhydroxy polymer and has been studied intensively due to its good thermal stability, biocompatibility, chemical resistance, hydrophilicity, biocompatibility, and biodegradability, and inexpensiveness [13,14,15,16].

Many biomedical products, including surgical sutures [17,18], contact lenses [19,20], and wound dressing [21], are made from PVA. It is also known that PVA can be employed in internal biomedical applications by using artificial organ designs such as artificial kidney membranes [22], and articular cartilage for orthopedic implants [7]. PVA has a wide range of uses in the textile industry [23], either as a formulating agent such as a thickener [24], surfactant [25], sizing agent [8,26], or microcapsule membrane material [27,28], or as coating [29], or water-soluble synthetic fiber [30]. PVA-based fibers have many potential applications in the textile industry. For example, short fibers are often used as reinforcement material in geotextile cements or cementitious composites [31,32,33], while long fibers are used in the manufacturing of geotextile products for soil reinforcement. PVA fibers have drawn tremendous interest for decades due to their unique physico-chemical and mechanical properties, which make them widely applied in various areas depending on the fiber mean diameter. Nanofibers are mainly used in applications such as membrane fabrications, tissue engineering, biomedical devices, and optical sensors [34]. Although its use in clothing is much less popular than that of other synthetic fibers, PVA fiber is still known for its moisture absorption and wear-resistance properties. The use of PVA multifilament yarns has been studied for biomedical applications [35], to improve the comfort of textiles [36,37], to design electrically conductive structures [38], as superabsorbent textile fabric [39], as sacrificial fibers for porous structures [40,41,42,43,44], and so on. The properties of PVA-based yarns vary according to the production method used and the intended application, such as wet spinning, melt spinning, gel spinning, and wet-dry spinning [45].

Electrospinning is a robust and versatile method for manufacturing fibers from the nanometer to the micrometer scale, based on adjusting a wide range of operational, process, and formulation parameters [46]. This method is based on establishing an electrostatic force to stretch polymer solutions from a Taylor cone formation to continuous fiber [47,48]. The conventional device consists of a high-voltage power supply, a syringe pump to regulate polymer solution flow through a capillary die or conductive needle, and the resulting fibers collected on a collector [49]. The significant variability in plant modifications and spinning solution formulations makes it possible to diversify the method. The electrospinning process involves many interrelated variables. Changing one of these parameters can influence the required range of another for an optimized process. The process parameters usually adjusted to control the physical properties of the fibers include the type of collector with different topographies, the distance between the polymer jet emission zone and the collector (tip to collector distance—TCD), the voltage applied to the power supply, the inner diameter of the nozzle or spinneret (gauge), and the flow rate of the spinning solution [50]. Fibers are obtained when the electrostatic force’s value exceeds the spinning solutions’ surface tension to form a Taylor cone [51]. It is, therefore, necessary to control the physico-chemical properties of the solutions (viscosity, surface tension, conductivity) related to the choice of polymer and its chemical characteristics (molecular weight, concentration, solubility) [52]. Environmental parameters such as relative humidity and temperature also influence the electrospinning process. Variations in relative humidity lead to changes in the solidification process of the charged jet, resulting in variations in nanofiber diameters and morphologies, mainly through the creation of pores when a mixture of solvents is used [53]. An increase in temperature modifies the solutions’ viscosity and the solvent’s evaporation kinetics, reducing the nanofibers’ diameter [54].

This review aims to provide an overview of PVA electrospinning, including principles, methods, PVA types, and applications. We begin by discussing the principle and typical electrospinning apparatus to provide readers a clear picture of this versatile technique. We then review the grades of PVA and formulations typically used for electrospinning. Next, we examine how formulation compositions influence the resulting morphologies designed to target different types of applications. All these attributes make PVA-based electrospun nanofibers a class of nanomaterials well suited to a wide range of applications, including tissue engineering [55], drug delivery [56], gas sensors [57], bioactive fibers [58], electronic skin or smart textiles [59], energy storage [60], and thermal comfort [61]. We focus on the most relevant examples to highlight the advances linked to developing PVA-based electrospun nanofibers.

## 2. Electrospinning of PVA-Based Membrane

The electrospinning method is an alternative approach that enables the production of fibers with diameters from a few nanometers to several micrometers and nonwoven surfaces composed of these fibers by forming an electrically charged polymer solution or melt jet [62]. Due to their high surface area/volume ratio, tiny pore diameters, and high porosity, electrospun ultrafine fiber surfaces are particularly appealing for usage in the medical industry [63]. Besides that, various fields such as food packaging [64,65] and air [66] and water filtering [67,68] can benefit from the use of nanofibers. Its shape and topographic features are similar to the extracellular matrix, so it is an ideal environment that can provide cell adhesion, proliferation, and differentiation and can be a carrier surface for bioactive active substances such as drugs and growth factors [69,70,71,72].

The electrospinning approach presents a simple procedure for producing fibrous surfaces with superior mechanical properties, substantial surface areas, and dimensions on the order of several nanometers [73]. Electrospinning is a fiber production method that uses a top-down engineering approach to draw ultra-thin fibers from a viscoelastic polymer solution or polymer melt using electrical force. A basic setup includes a syringe with a needle, an injection pump for the spinning solution, a voltage power supply, and a collector at an optimum distance [74]. In principle, four successive processes produce the fiber jet from the polymer droplet (Figure 1). Positive charges gather around the droplet due to the high voltage supplied to the polymer melt or solution at the syringe tip. As the voltage rises, the surface tension of the polymer at the nozzle tip cannot resist the repelling force of the charges, and the droplet turns into a conical shape known as a Taylor cone. The droplet carries fewer positive charges than the polymer jet after developing the Taylor cone. A continuous fiber jet is ejected from the Taylor cone’s tip toward the collector at a critical voltage. The density of the surface charges rises as the fluid jet moves closer to the collector. Surface charges eventually prevail over surface tension, causing this continuous thin fiber to begin coiling. A fibrous, nonwoven surface is formed by the piling of the solid fibers reaching the collector after the complete evaporation of the solvent during the jet’s ejection to the collector [75,76].

Numerous equipment configurations have been created to improve the typical electrospinning plant, overcome various plant limitations, improve the spinning process, further improve in versatility, and perform the resulting fibrous materials. The classification shown in Figure 2 is divided into two leading families based on the use or non-use of needles and considers the method of fiber generation, die movement, and fiber collection direction, as well as the use or non-use of needles [78]. Methods based on using multiple jets have been created to improve nanofiber production. However, because of the multiple jets, the nanofiber web is not uniform due to the replenishment effect between the jets. Needle-free processes have been developed to improve nanofiber production rates and depend, in this case, on the dies’ shape, which also modifies morphology [79].

The device is typically horizontal [92], but can alternatively be vertical [80,93,94,95]. The resulting fiber morphology and spinning efficiency differ in horizontal and vertical systems due to gravity’s impact on the Taylor cone and polymer droplet morphologies [76,96]. Although static collectors are frequently employed in the literature [97,98], dynamic collectors like rotating drums [94,99] and rotating discs are also available [100], which create a more aligned deposition (Figure 3) [101].

The optimization of electrospun fiber is based on the final application and researchers’ desire. However, it should be known that six forces need to be considered for optimizing the parameters in the electrospinning method, i.e., (i) the gravitational force; (ii) the electrostatic force exerted on the charges present along the polymer jet, and allowing its movement from the needle to the collector; (iii) the repulsion forces between charges of mutual polarities or coulombic stretching force that allows the thinning or stretching of the charged jet during its flight toward the collector; (iv) the viscoelastic force that attempts to prevent the charged jet from stretching; (v) the surface tension that opposes the stretching of the charged jet’s surface; and (vi) the drag force initiated by the friction between the charged jet and the environment [13]. The influence of the force of gravity on the electrospinning process, which depends on the density of the solution, depends on the configuration of the equipment. Apart from the fact that it affects the formation of the Taylor cone and the shape of the droplet leaving the nozzle, its existence in a horizontal process will require higher electric fields to overcome it, as well as the surface tension of the solution. In the case of a vertical installation, they will act on the jet of polymer solution, accelerating its elongation. Furthermore, it appears that the influence of this force, like that of electronic forces (a function of the electric field and the conductivity of the solution) and drag forces, is limited on the formation and morphology of nanofibers compared with the coulombic, viscoelastic, and surface tension forces [102].

The optimizing of the conditions of the electrospinning process results in homogeneous, reproducible electrospun nanofibers. The primary known defects occurring in the electrospinning process are the presence of branched nanofibers, beaded nanofibers, curled nanofibers, and flat (ribbon-like) nanofibers. Thus, process, formulation, and ambient parameters affect the homogeneity and reproducibility of the electrospinning process. Parameters affecting the electrospinning process can be divided into three categories, i.e., (i) process parameters, (ii) solution parameters, and (iii) ambient parameters. Although the optimization of these three categories of parameters is correlated, each has a distinct effect on fiber morphology and must be studied to produce electrospun fibers with the desired morphologies and diameters [103].

### 2.1. Influence of Operating Parameters on Electrospinning PVA Nanofibers

#### 2.1.1. Voltage

The electric field strength is known to increase with an increase in the applied voltage. However, a critical voltage is required to obtain the cone for fiber formation, even though the working distance is minimal with a high electric field strength. The unbalanced effect results in the prepared fiber exhibiting a wide size distribution when high voltage is applied [104].

The applied electrical voltage has an evident influence on the fiber’s morphology. The surface tension of the solution and the electrostatic force should be balanced to obtain stable jets to form continuous fibers. Once the applied voltage exceeds the critical voltage, the jets of the liquids will be ejected from the cone tip. The jets, responsible for the bead formation (sometimes called electrospray if the beads are separate particles), will not be stable if the solution viscosity is extremely low. High voltages can generate more charges to the solution or droplet surface located at the tip of the needle (higher coulombic forces) as well as a higher electrical field (larger electrostatic forces), both of which will stretch the jets fully for the favorable formation of uniform and smooth fibers [105,106].

Ding et al. observed that the fiber diameters decreased with an increase in the applied electrical voltage [107]. Also, they achieved fibers with more uniform sizes at a higher applied electrical voltage (20 kV). The increase in the electric field strength should enhance the electrostatic force on the solution jet, which will facilitate thinner fiber formations. In another study, increasing the voltage from 15 to 30 kV decreased the mean diameter of nanofibers by 25% [108].

#### 2.1.2. Feed Rate

A low feed rate results in the presence of a vacuum inside the needle; however, larger feed rates cause the polymer to deposit on the tip’s edge, which disrupts the Taylor cone formation. To achieve and maintain a stable Taylor cone for each applied voltage, an ideal feed rate value is required [109,110].

It has been pointed out that the polymer solution’s feed rate strongly influences the polymer fiber morphology [34]. There is an optimum for the feed rate, and once the feed rate is sufficient to form fibers, a higher feed rate will supply more PVA solution, which is recognized as excess, producing fibers with beads [105]. Ding et al. observed that the average diameter was almost the same with an increasing feed rate [107]. This indicates that the electrospun PVA fibers’ morphology, diameter, size, and uniformity are not significantly influenced by the feed rate.

#### 2.1.3. Tip-to-Collector Working Distance (TCD)

The TCD primarily influences the evaporation of the solvent in the solution and the stretching of a PVA macromolecular chain, but it also affects the strength of the electric field. If the TCD decreases, the strength of the electric field increases. Polymer jets are stretched under electrostatic force when they travel through the electrical field. During this process, the solvent evaporates, and the polymer jets maintain their elasticity. It is reported that the possibility of fiber contraction (shrinkage) increases with an increase in the working distance [108]. If the fiber diameter is observed to decrease, uniformity is achieved by increasing it gradually with increases in the working distance to reach an optimum point; after this point, by increasing the distance, the uniformity of the fibers decreases, and even some cracks are observed in the PVA fibers [107]. However, varying the TCD over a small distance range, between 10 and 15 cm in the case of Phachamud’s experiment, did not limit the formation of pearled fibers or defects [104].

The influence of increasing the distance on the average diameter of nanofibers is a controversial topic in the literature. On the one hand, Ding et al. observed that the diameter of a PVA nanofiber slightly increases by increasing the tip-to-collector distance from 6 to 14 cm [111]. On the other hand, Supaphol and Chuangchote attributed the decrease in fiber diameter to the increase in distance, resulting in an increase in the total trajectory of the charged jet, time of flight, and more uniform stretching [13]. Selected SEM images in Figure 4 illustrate the morphological appearance of the as-spun fibers from the sonicated 10% *w*/*v* PVA solution collected at various collection distances in the range of 5 to 20 cm. Evidently, at the collection distance of 5 cm, a combination of smooth and beaded fibers was obtained, and some of the adjacent fibers appeared to fuse at touching points, an indication of the incomplete drying of the jet prior to deposition on the collector. Increasing the distance between the needle tip and the collector reduces the electric field intensity, decreasing electrostatic and coulombic repulsion forces. These changes lead to instability in the emission of the solution jet and, consequently, to an increase in the flight path, allowing the solvent to evaporate and the stretching of the polymer jets to result in fibers with smaller diameters. Conversely, a decrease in the TCD generally induces incomplete solvent evaporation, with partially solvent-swollen fibers collapsing to form non-uniform fibers with larger diameters or beads. Thus, increasing the trajectory between the needle point and the collector is necessary for obtaining smooth, uniform fibers or microparticles.

#### 2.1.4. Collector Design

Conductive collectors act as a substrate to collect the charged fibers during the electrospinning process. The kind of collector majorly influences the structure and performance of the collected nanofibers [112]. Due to the instability of the electrospinning jet, typical electrospun nanofibers are generally collected as non-woven or randomly oriented structures, especially when using a static collector such as a metal plate. Only coulomb forces produced by an applied voltage act on the electrospinning process in the static collector. As the fibrous structure’s disordered orientation and low mechanical strength have limited applications, many studies have focused on fiber alignment [113]. Furthermore, it can be noticed that nanofibers with a similar alignment and average diameter also have similar melting temperatures and crystallinity [114].

One method of obtaining aligned electrospun fibers is to use parallel electrodes, or two parallel plates (separated by a gap) connected to a grounded electrode [115]. As the distance between the two conductive collectors increases, the average diameter of the nanofibers decreases with a high degree of alignment [116]. The concept proposed by Rakesh et al. is based on modifying the nature of the electrostatic field on the collector plate by creating a repulsive field of different magnitudes and directions using collector plates with various geometric slots to produce aligned electrospun nanofibers [117]. A modification of the frame collector technique, known as the dual vertical wire technique, was introduced by Chuang-chote and Supaphol [118]. This approach involves two vertical stainless steel wires as the secondary target and a grounded aluminum foil as the primary target. The wires are positioned in parallel along a central axis between a blunt-ended stainless steel needle and the grounded aluminum foil. Both the needle and the foil are tilted at approximately 45 degrees from the vertical baseline. This configuration enables the simultaneous collection of aligned fibers (between the parallel wires) and a randomly aligned fiber mat (on the aluminum foil).

The rotating collector, in addition to coulomb forces, can offer a mechanical stretch force to align molecule chains. In other words, during electrospinning, the high-speed rotating collector supplies elongation forces and arranges macromolecular chains to produce aligned fibers. The rotary drum is the most widely used collector, where the fiber diameter is controlled by varying the drum speed. Fiber alignment is achieved with an improved mechanical performance at higher speeds [119,120]. Li et al. designed a novel tubular collector with helically arranged metal wires for electrospinning PVA-styrylpyridinium pendent groups (SbQ) fibers into a 3D helical structure [121]. This method not only offers the potential for creating biomimetic scaffolds or grafts, but also provides an approach that could lead to the development of scaffolds more precisely mimicking the fibrous structure of specific blood vessels. Using a rotating disc represents a significant advantage over other rotating systems, as the edge of the disc is covered by a massive deposit of aligned fibers [122]. An electric field exists between the polymer jet and the rotating wheel, and as the wheel rotates, the field is affected by the sharp edge of the wheel. In addition, the polymer jet is subjected to a tensile force from the wheel, causing the deposited nanofiber to stretch and become finer. The deposited fibers align themselves while remaining separated using a repulsive force, as each of the fibers carry a residual charge as it accumulates in the rotating collector. Another approach was developed by Chvojka et al. [123], who used a special saw-shaped collector to produce PVA nanowires. The collector shape was designed to align the nanofibers in the space between neighboring lamellae, thanks to the distribution of the electric field in the vicinity of the collection device.

### 2.2. Influence of Formulation Parameters on Electrospinning PVA Nanofibers

The effects of processing parameters for the polymer solution and processing condition on the morphology, such as the polymer concentration, its molecular weight, and the solution’s electrical conductivity, were found as dominant parameters to control the morphology of electrospun polymer fibers [124]. The diameter of the electrospun fibers dramatically decrease with a decreasing polymer concentration. Surface tension effects could dominate with a decreased polymer concentration and solution viscosity, and beaded fibers are consequently produced. Non-uniform fibers are formed more easily with a low polymer molecular weight. The molecular weight of the polymer reflects the number of entanglements of polymer chains in a solution. Thus, sufficient solution viscosity is necessary to produce a uniform jet during electrospinning and restrain surface tension effects. It was found that the diameter of the electrospun fibers is not dramatically changed with varied applied voltages. The voltage effect is notably diminished when the polymer concentration is low.

#### 2.2.1. Polymer Concentration, Viscosity, and Surface Tension

The molecular weight (Mw) of commercially available PVA is generally between 9000 and 186,000 g.mol^−1^ [125]. PVA solution viscosity increases with concentration, impacting the electrospinning process. Excessively viscous solutions can lead to beaded fiber formations, while insufficient viscosity prevents jet development and fiber growth. Colby et al. identified four concentration regimes, i.e., (i) diluted, (ii) semi-dilute unentangled, (iii) semi-dilute entangled, and (iv) concentrated [126]. The critical concentration (C*) transitions to the semi-dilute unentangled regime, where beaded fibers form. The entanglement concentration (Ce) signifies a significant viscosity increase, ensuring consistent bead-free fibers (Figure 5). Highly viscous solutions can cause issues during feeding [127,128]. The critical concentration (C*) can be determined as 1/[η], and the transition from semi-dilute untangled to entangled regimes is determined from viscosity vs. shear rate graphs. Gupta et al. [12] investigated various c/C* values for different solution regimes, including dilute (c/C* < 1), semi-dilute unentangled (1 < c/C* < 3), semi-dilute entangled (3 < c/C* < 6), entanglement concentration (Ce) (c/C* > 6), and highly concentrated solutions (C** = c/C* > 10) [129]. Above C**, fibers become significantly concentrated, increasing in diameter and losing their circular cross-section. Notably, the theoretical crossover value of semi-dilute unentangled to entangled regimes (C*) is higher than the experimentally determined value.

Polymer concentration is an essential operational parameter in the electrospinning process, significantly influencing the fiber morphology. The formation of the fibers is inhibited in the solutions with high concentrations due to their high viscosity [105]. The high viscosity makes it extremely difficult for the solutions to flow through the syringe needle to form nanofibers under an electrostatic force [104]. Therefore, an appropriate solution concentration becomes one of the critical parameters required to optimize the final electrospinning fibers. An extremely low solution viscosity can cause the formation of beads in nanofibers. In a solution with relatively high viscosity, the stable jets, without breaking due to the cohesive nature of the high viscosity, travel to the electrode and finally form the uniform fibers on the collecting grounded electrode. The viscosity and conductivity of the PVA solution are evidently increased as the polymer concentration was increased [104]. In another study, Supaphol et al. reported increased viscosity and conductivity by increasing the PVA concentration [13]. By increasing the viscosity, the average diameter of the PVA nanofibers increases by 18 and 36% with an increase in polymer concentration from 9 to 10 and 11% by weight [107]. This means that a high-concentration solution results in high viscosity and high surface tension; after that, the stretching ability is reduced, so the diameter of the nanofibers increases. In a study by El-Aziz et al., they chose 10 wt.% of PVA water solution based on the smallest fiber diameters and the highest surface area of the membrane [108]. Another item that affects the viscosity of the solution is the use of sonication for PVA. Using sonication to prepare a PVA solution causes the viscosity of the solution to decrease the average diameters of the as-spun fibers [13].

Solution surface tension plays an essential role in electrospinning since it is the main force opposing the repulsive force of the charges on the jet surface. As such, it influences the formation of the Taylor cone and the initiation of the solution jet and, consequently, the fiber diameter [130]. In addition, a high surface tension has much the same effect as low viscosity, i.e., the creation of instabilities in the solution jet, leading to the appearance of beaded filaments. If the surface tension is too high, a higher voltage must be applied to produce fibers [131]; if the surface tension is too low, it is no longer suitable.

This characteristic essentially depends on the composition of the polymer solution, i.e., the choice of solvent for solubilizing the PVA, the physico-chemical properties of the PVA (molecular weight (g.mol^−1^) (Mw), degree of hydrolysis (DH), concentration (wt.%)) [132,133,134], and the presence or absence of a surfactant [135]. PVA, as a polymer colloid, has a surface effect in an aqueous solution by creating H-bonds between the hydroxyl groups of the PVA chains and water. At low concentrations, the surface tension of an aqueous solution decreases drastically, reaching a plateau between 40 and 50 mN/m depending on the formulation used, for a polymer concentration range from 0.1 to 0.2 wt.%. The decrease in surface tension is linked to the adsorption of macromolecules at the air–water interface as concentration increases. For the concentration ranges used in electrospinning, the surface tension of polymer solutions does not change, regardless of the polymer concentration and molecular weight used, indicating solution saturation [136].

To reduce the surface tension of PVA-based solutions, in addition to selecting solvents other than water, a co-solvent with low surface tension leads to smooth nanofibers [137]. Another strategy described in the literature is to use a surfactant [135,138], which also leads to uniform nanofibers. The introduction of non-ionic and anionic surfactants at low concentrations contributes to a reduction in solution surface tension in PVA solutions. Indeed, for concentrations above 1% (*v*/*v*) or above the critical micelle concentration, it has been observed that surface tension increases due to interactions between polymer and surfactant, and to the formation of free micelles corresponding to the saturation of polymer–surfactant interactions. Adding an ionic surfactant, such as sodium dodecylbenzene sulfonate (SDBS) not only decreases the surface tension of the solution, but also helps increase electrical conductivity, thereby reducing the nanofiber diameter and contributing to fiber homogeneity [138]. Jia and Qin observed that the introduction of cationic surfactants led to a decrease in surface tension by 33%, depending on the concentration [135]. When the concentration of anionic surfactant was less than 1% (*v*/*v*), the surface tension of the PVA solution with anionic surfactant was reduced to half; whereas with 1.2% (*v*/*v*) anionic surfactant, it increased by half. The addition of non-ionic surfactant decreased the surface tension to 4.6 mN/m; and similar to the introduction of anionic surfactant, when the concentration of the non-ionic surfactant was 1.2% (*v*/*v*), the surface tension increased to 32.6 mN/m. On the other hand, the effect of using an amphoteric surfactant on surface tension was negligible. These variations in surface tension due to macromolecule–molecule interactions are also correlated with variations in viscosity in the dope solution.

#### 2.2.2. Conductivity

The conductivity of a fluid has a direct impact on the amplitude of jet instability modes. A solution with a high conductivity has a greater capacity to transport charges than a solution with a low conductivity, which increases the stretching of the solution jet. This parameter plays a less important role than the other physico-chemical parameters of solutions. Highly conductive solutions cause jet instabilities in the presence of high electric fields, resulting in instabilities at the Taylor cone and consequently increasing the mean diameter and distribution of the resulting nanofibers [139]. The solutions’ conductivity also influences the nanofibers’ morphology, and Itoh et al. observed that the introduction of salt led to an increase in conductivity, resulting in the development of ribbon-like fibers [140]. However, the effect of salts on the nanofibers’ morphology and average diameter also depends on these species’ chemical nature [134] and their interactions with macromolecules, which influence all other physico-chemical properties.

#### 2.2.3. Solvent

Solvents influence the physico-chemical properties of solutions, particularly surface tension. A high concentration of free solvent molecules contributes to the aggregation of solvent molecules, enabling the solution to adopt a spherical shape. Interactions between solvent and macromolecules increase viscosity, so when the solution is stretched under the influence of charges, the solvent molecules will tend to spread out over the entangled macromolecules, limiting the aggregation of solvent molecules under the influence of surface tension [135].

The physico-chemical properties of the solutions also depend on the solvent used to solubilize the PVA. Solubility also depends on the degree of polymerization, hydrolysis, and solution temperature [141]. A change in any of these factors alters the possibility of hydrogen creation in the medium, affecting solubility and solution properties. PVA is soluble in polar solvents such as water, dimethyl sulfoxide (DMSO), ethylene glycol (EG), and N-methyl pyrrolidone (NMP) [142], though water is the most commonly used solvent for PVA. The use of a solvent mixture has been reported several times in the literature. However, it also appears that a mixture of two suitable solvents, such as water and DMSO, acts as a non-solvent for PVA due to stronger solvent–solvent interactions than polymer–solvent interactions [143]. Therefore, the morphology of electrospun nanofibers depends on the ratio between the two solvents, water/DMSO. According to Gupta et al., low DMSO content leads to uniform fibers, with an increase in diameter as a function of DMSO addition [142]. This variation is linked to a change in the rheological properties of the solutions and, in particular, the relaxation time of the macromolecular chains. The solutions exhibit poor spinnability at molar ratios of 1:2–1:3, forming non-uniform pearled fibers. The choice of a co-solvent can also be made based on a Teas graph, considering the solvents’ polar, dispersive, or hydrogen components. For example, Mahmud et al. found that solution spinnability improved with increasing dispersion strengths or dispersive components for water–ethanol or water–acetic acid mixtures, with a contribution to solution viscosity, particularly for high-molecular-weight PVA [144]. During the process, solvent evaporation kinetics determine the nanofibers obtained. Adding a low-vapor-pressure solvent such as dimethylformamide (DMF) to the aqueous solution provides greater control over fiber morphology [145].

### 2.3. Influence of Relative Humidity on Electrospinning PVA Nanofibers

Relative humidity influences solvent volatilization, which influences PVA nanofiber formation and fiber surface structure or morphology. Depending on the relative humidity, the electrospun nanofiber’s diameter changes. A thicker nanofiber results from quick solvent evaporation under low relative humidity, while a thinner nanofiber results from increasing the relative humidity’s inhibition of solvent evaporation, because high evaporation increases the spinning fluid viscosity, and polymer chains are not subjected to voltage-induced stress, increasing the fiber diameter [146,147,148]. Due to PVA’s great hydrophilicity, the environment’s humidity will impact how nanofibers develop and the physical characteristics of the resulting structures. Only a few articles have focused on the effect of moisture on PVA nanofibers. The impact of moisture on the morphological and mechanical characteristics of PVA nanofibers was established by Pelipenko et al. [147]. The diameter of the nanofibers decreased with increasing relative humidity (RH%), from 667 ± 83 nm (RH 4%) to 161 ± 42 nm (RH 60%). The relative standard deviation significantly increased, and uneven fiber morphology with beads was observed at the most significant relative humidity tested (70%). Additionally, mechanical analyses of nanofibers below 250 nm showed that the stiffer fiber structures resulted from the reduction in nanofiber diameter produced with higher RH values employed during electrospinning.

Nanofibers with diameters greater than 250 nm have constant mechanical characteristics due to size-dependent surface effects. Changes in RH substantially influence the packing density and fiber diameter [146]. Higher RH values support the construction of a more closed structure with thinner diameters, whereas lower RH values favor an open structure. This is explained by two phenomena that happen when RH increases: the polarization of the air caused by the high electric field created by the formation of ozone and the increase in conductivity of the medium as a result of the increase in conductive paths created by the water dipoles along the fiber spinning gap. Raksa et al. investigated the effects of RH on surface porosity and fiber diameter on PVA-silk fibroin blend nanofibers [148]. A less interconnected pore distribution was found on the surface in low RH conditions since the rapid solid fiber formation due to solvent evaporation limited the entrapment of water molecules and condensing of moisture on the surface during the process. The nanofiber electrospun at the lowest RH (50%) has the maximum tensile strength with the lowest % elongation at break. The nanofibers displayed a loose texture at a high RH (80%), indicating inadequate fiber–fiber bonding.

## 3. Morphology of Electrosprayed/Electrospun PVA Materials

The properties of electrospun PVA are greatly influenced by its Mw and DH, impacting physical and chemical characteristics, working concentration, fiber morphology, cross-section, and the occurrence of bead formation [93]. Lee et al. found that a higher Mw PVA produces superior physical characteristics, enhancing the electrospun mats’ stability and mechanical strength [94]. Peresin et al. noted a significant increase in the degree of crystallinity of electrospun PVA due to improved crystallization under high shear stresses, though the polymer matrix’s structure remained unaffected [149]. Additionally, Restrepo et al. reported that polylactid acid/PVA (PLA/PVA) blended with higher Mw and PVA hydrolysis levels exhibited superior thermal stability [150]. Considering the multi-disciplinary nature of PVA, it is notable that limited research has explicitly investigated the impact of physico-chemical parameters within PVA solutions on the fabrication of electrospun structures. Understanding these relationships between Mw and DH within PVA solutions is crucial for optimizing electrospun mats, particularly for applications such as wound dressing, tissue engineering, and drug delivery systems.

Low polymer concentrations (c < C*) correspond to a dilute regime where the polymer chains are isolated. The critical concentration C* marks the boundary between the dilute and semi-dilute regimes. There is no cohesion in the polymer chains in this state because they are too far apart to interact. The Rayleigh instability will induce droplet formation, and only the particles will be collected. This process is called electrospraying [151,152,153,154]. Electrospraying is a unique method for creating polymer-based micro- and nanoparticles. It has recently gained popularity due to its benefits, including ease of batch manufacturing, high yield, and narrow size distribution [155]. The electrospraying technique is based on the same principle as the electric field fiber-spinning method. The difference between the two techniques is achieved by varying the properties of the solution, such as concentration, solvent, and viscosity, and process parameters, such as flow rate, distance from needle tip to collector, and voltage [156]. Micro- and nanoparticles can be formed by electrospraying, disrupting the Taylor cone and endless fiber formation stage and forming droplets (Figure 6) [72]. Li et al. encapsulated in situ synthesized Fe_3_O_4_ nanoparticles inside a PVA shell by electrospraying for magnetic resonance imaging, which has been developed as an alternative to X-ray digital subtraction angiography. Additionally, Fe_3_O_4_ nanoparticles served as physical crosslinkers to help the PVA droplets gel and solidify into stable PVA matrices [155]. The size distribution of nanoparticles with morphology close to spherical was between 262 µm and 958 µm. Young et al. encapsulated L929 fibroblast cells within PVA microspheres [157]. Unlike in the general approach, microcapsules were atomized into a liquid rather than a surface, and this method is called submerged electrospraying. The microspheres atomized into hydrogel macromer solution were simultaneously covalently crosslinked under UV light. This method successfully produced uniform, spherical particles with a customizable mean size from a few micrometers to several hundred micrometers, depending on the flow rate and applied voltage.

The viscosity of PVA solutions is directly related to PVA concentration, impacting entanglement. As PVA concentration increases, viscosity rises (Figure 5). Distinct slopes in the viscosity–concentration curve mark the transition from diluted to semi-diluted regimes. Increased concentration leads to greater entanglement. Concentrations above the entanglement concentration (Ce) have been used for electrospinning experiments. Viscosity influences the formation and stability of the polymer jet and resulting mat. Morphologies of electrospun PVA mats were compared with those in the literature based on Mw and Berry number (Be = [η]C) (Figure 7). Be indicates chain entanglements, with Be > 1 denoting entanglement. Be values between 4 and 9 result in beaded fibers, while a Be around 9 leads to beadless fibers [158]. In our study, beaded fibers were observed at Be values between 4.6 and 8.6. Beadless nanofibers were achieved with a higher Mw PVA (Mw 31,000 and 61,000 g.mol^−1^) at Be values of 7.2 to 10.8. To reduce Ce for a high Mw PVA (130,000 g.mol^−1^), a co-solvent such as acetic acid (HAc) can be introduced, increasing electrical conductivity [159]. However, higher conductivity may lead to an unstable cone jet mode due to coulomb repulsion [160]. A bead-free nanofibrous surface was achieved with 5% PVA-5 (7/3 distilled water/HAc) at Be = 5.3, which is lower than those of beadless nanofibers produced with water alone (Figure 7).

Viscosity is important in fiber formation and morphology, directly influenced by solution concentration and polymer Mw. Figure 8 illustrates the dependence of PVA fiber morphology on Mw and concentration. The data do not exhibit distinct groupings. Low Mw PVA (9000–10,000 g.mol^−1^) failed to produce beadless fibers even at a high viscosity (776 mPa.s). A viscosity of 399 mPa.s or higher is required to achieve non-beaded fibers, particularly for medium and high Mw PVA (Mw > 61,000 g.mol^−1^). However, it is essential to consider the DH. PVA with Mw: 9000–10,000 g.mol^−1^ has the lowest DH (80%), indicating a higher proportion of vinyl acetate monomers. The presence of acetyl groups in PVA can impact hydroxyl group interactions, potentially altering electrospinning conditions due to reduced hydrophilicity. The findings on the orientation show that regardless of various production parameters and PVA types, there is no significant variation in fiber alignment on the resulting web-like PVA surfaces.

## 4. Applications of PVA Nanofibers

Electrospun nanofibers have been employed industrially for more than two decades. Synthetic polymers have been focused on because of their inexpensiveness, abundance, higher chemical and thermal stability, and more accessible and uniform fiber production [162,163]. The efficiency and reproducibility of the electrospinning process of proteins and polysaccharides, as well as the fiber uniformity, remain problematic due to the highly diverse chain conformations, hydrodynamic responses, and repulsive forces in solution among the polyanions of natural polymers, which restrict their practical application [164]. The desired application benefits from increased mechanical and biological features when blended with natural polymers. PVA is a superior carrier polymer for blending natural polymers due to its low cytotoxicity, water-solubility, and relatively more straightforward electrospinnability. PVA can decrease repulsive forces in charged biopolymer liquids, enabling fibers to electrospin [163,165]. The temperatures at which PVA dissolves span a considerable range. PVA dissolves partially at low temperatures, but entirely at temperatures above 70 °C, depending on the molecular weight and degree of hydrolysis [166]. Another benefit of employing PVA as a hydrophilic component in blends is that, at body or room temperature, it does not quickly dissolve and leach into water or culture media [167].

Electrospun ultrafine fiber surfaces are particularly appealing for medical and cosmetic purposes because of their high surface area/volume ratio, small pore sizes, and high porosity. It has a perfect environment for cell adhesion, proliferation, and differentiation and can be a carrier surface for bioactive active substances such as medicines and growth factors due to its structure and topological properties that are similar to those of an extracellular matrix [69,70,71,72].

### 4.1. Filtration

Filtration is a fundamental application area where electrospun nanofibers can be used because of their high porosity, interconnected open pore structure, and desirable membrane thickness. Maleic acid-crosslinked PVA nanofiber layers have been employed as a sub-layer on spunbond or meltblown surfaces for air filtration [163,168], developed PVA/chitosan (PVA/CS) and PVA/cyanobacterial extracellular polymeric substances (EPS), and blended nanofibrous membranes for water filtration. The article showed the capability of blending EPS with a PVA blend for the first time while having higher mechanical properties with a lower disintegration in between 10 and 50 °C and a superior chromium binding capacity than PVA-CS blends. The use of PVA in combination with other materials, such as chitosan, has been reported to enhance the antibacterial properties and filter efficiency of the nanofibrous membranes, making them suitable for filtration applications [169]. Additionally, the incorporation of photocatalytic materials, such as titanium dioxide (TiO_2_), into PVA nanofibers has been shown to improve filtration efficiency, highlighting the versatility of PVA-based nanofibrous membranes for filtration studies [170].

In the time of the COVID-19 pandemic, the significance of personal protective equipment (PPE) heightened. Electrospun nanofibers, particularly those incorporating PVA, have emerged as versatile materials for advanced PPE development. Numerous studies have investigated the use of electrospun nanofibrous membranes in PPE applications, showcasing their potential for providing UV-shielding, antibacterial and antiviral properties, and high filtration efficiency [171,172,173,174]. Abbas et al. developed a novel multilayer mask with a degradable, multifunctional hybrid filter composite. This composite, comprising three electrospun nanofibrous layers, addresses challenges in face mask filters. The outer layer, a TiO_2_/CS/PVA matrix with TiO_2_ nanotubes, acts as an antimicrobial and antiviral agent. The middle CS/PVA layer provides natural air filtration and pathogen inactivation. The inner layer, composed of silk/PVA nanofibers, enhances mechanical properties and heat dissipation, contributing to wearer comfort [175]. Moreover, the combination of PVA with other materials, such as metal–organic frameworks, has been employed to construct composite membranes for highly efficient air filtration [176].

PVA has also been explored for chemical filtration applications. Amino group (Hal-NH_2_)-grafted halloysite nanotubes were utilized to enhance adsorption/filtration in electrospun PVA/CS nanofibers. Employed in two modes—coated on the membrane and embedded in the polymer solution—these cactus-like nanofibers demonstrated significantly higher divalent cadmium (Cd(II)) and divalent lead (Pb(II)) adsorption capacities compared to alternative functionalized materials. Notably, the stability of Hal-NH_2_ entrapped nanofibers endured, maintaining a nearly intact adsorption capacity over five adsorption/desorption cycles, in contrast to Hal-NH_2_ coated samples [177].

Yang et al. have developed a green and environmentally friendly material for disposable protective products such as masks [178]. The fiber membranes were modified with glutaraldehyde (GA) vapor and hydrochloric acid to improve their resistance to hydrolysis. The fiber membrane inhibited *Staphylococcus aureus* and *Escherichia coli* by over 97%. The fibers exhibit free aldehyde groups after GA crosslinking, which have good antibacterial properties. The filtration efficiency of the CS/PVA/GA fiber membrane was over 95%, and the filtration resistance did not exceed 343.2 Pa, meeting the requirements of the filter material used in the mask. The use of GA/HCl (hydrochloric acid) vapor for chemical crosslinking resolves the instability of electrospun PVA fibers in an aqueous environment. The CS/PVA/GA nanofiber composite membrane offers good hydrolysis resistance while retaining good antibacterial properties and filtration efficiency.

### 4.2. Gas Sensor

Metal oxide semiconductors are a precious class of materials due to their potential applications in photocatalysis, solar cells, and gas sensors. In contrast to traditional synthesis methods, such as vacuum deposition, spin-coating, sputtering, sol-gel synthesis, and thermal evaporation, electrospinning offers a cost-effective and versatile option for the large-scale production of one-dimensional composite materials with high surface areas. The self-supporting, entangled fiber structure of the material produced through electrospinning also leads to high porosity, which is beneficial for gas-sensing applications. In reducing the size of the sensing material, the gas-sensing performance is increased due to the increased surface-to-volume ratio and increased surface reaction sites for the absorption of gas species [179].

Common polymers used in the production of electrospun metal oxide nanofibers include water-soluble polymers such as poly(vinylpyrrolidone), polyethylene oxide, poly(vinyl acetate), and PVA. The integration of multifunctional materials, including metal oxides, inorganic non-metals, metal–organic frameworks, and covalent organic frameworks, into these polymers allows for the creation of composite nanofibers with diverse morphologies, structures, and functionalities through controlled conditions [180].

Electrospinning is used due to several key advantages such as (1) the ability to produce nanofibrous membranes with a range of specific surface areas by controlling fiber geometry and pore size, which makes them ideal for use as sensors; (2) the versatility of fiber materials, which allows for a wide range of applications based on their unique characteristics; (3) the efficient molecular recognition capabilities of electrospun sensors, enabling rapid and simple detection without the need for the pre-treatment of samples; (4) the high sensitivity and quick response of the sensor interface to even small quantities of samples; (5) the excellent repeatability, stability, and reusability of electrospun sensors; and (6) their cost-effectiveness for detection. Furthermore, the large surface area of electrospun nanofibers provides ample active adsorption sites for target analytes, leading to accelerated adsorption and desorption rates for improved sensitivity and faster sensing [181].

Two significant methods exist for preparing sensors based on metal oxide materials using electrospinning. The first involves creating nanofibers with inherent sensing capabilities through the electrospinning of functional polymers, PVA, that serve as the sensors’ sensing elements [182]. The second method involves utilizing electrospun nanofibers as templates and depositing responsive sensing materials with surface functionalization to form micro- and nanostructures with specific sensing properties [183].

### 4.3. Biosensors

In the field of biosensors, the investigation of employing PVA nanofibers for immobilizing different nanoparticles, such as gold and palladium, has been conducted. This integration is specifically engineered to enhance the electrochemical activity and might provide antibacterial properties to the nanofibers. Xu et al. incorporated gold nanoparticles into electrospun PVA/polyethyleneimine (PEI) nanofibers. The glutaraldehyde-crosslinked nanofibrous mats served as nanoreactors, facilitating the complexation of tetrachloroaurate ion (AuCl_4_) anions by binding with the free amine groups of PEI. This process led to the subsequent formation and immobilization of gold nanoparticles (AuNPs), offering potential applications in catalytic activity and reusability for the transformation of 4-nitrophenol to 4-aminophenol [184]. Wang et al. proposed an efficient method for fabricating electrochemical biosensors, involving the immobilization of palladium nanoparticles onto PVA/PEI nanofibers through a combination of electrospinning and in situ reduction processes [185]. This approach holds promise for environmental pollution monitoring devices. The resulting nanocomposites demonstrated outstanding performances as electrochemical biosensors, enhancing the detection of H_2_O_2_ by significantly improving the electron transfer between the redox-active site of H_2_O_2_ and the glassy carbon electrode. Itani et al. developed a biosensor—a fluorometric electrospun fiber sensor mesh based on enzymes—using blends of polycaprolactone (PCL), poly(lactid-co-glycolic acid) (PLGA), and PVA nanofibers [186]. The resulting nanofibrous mat, produced with polymers, alcohol dehydrogenase, and oxidized nicotinamide adenine dinucleotide, successfully targeted gaseous ethanol.

### 4.4. Tissue Engineering

Tissue engineering is a multidisciplinary field that aims to develop biological substitutes to restore, repair, or replace damaged tissues and organs [187]. This field differs from wound dressing in that it involves the construction of cells, biomaterials, and bioreactors to develop three-dimensional artificial tissues and organs, whereas wound dressing is primarily focused on promoting wound healing and preventing infections [188]. The main organs of concern in tissue engineering include the bladder, blood vessels, heart valves, tendons, and ligaments, among others [189,190,191,192]. The tissue engineering triad consists of three main factors: cells, signaling molecules, and scaffold, which support and rely upon one another [193,194].

The significance of electrospinning polyvinyl alcohol (PVA) in tissue engineering is notable, illustrated by its diverse applications and contributions to the field. The extensive use of electrospun PVA nanofibers in tissue engineering is due to their biocompatibility, biodegradability, and adjustable mechanical properties, rendering them suitable for constructing scaffolds that mimic the extracellular matrix. This supports a conducive environment for cell attachment, proliferation, and tissue regeneration [121,195]. Koosha and Mirzadeh used PVA as a guest polymer to make chitosan electrospinnable by leveraging favorable interactions between these polymers [99]. Nanofibers were drawn from blends of different ratios prepared from a 7 wt.% solution of both polymers. While nanofibers could not be obtained from chitosan solution alone, it was revealed that solutions were more complex to spin when the ratio of chitosan to PVA was higher than 50/50. While the average diameter of PVA fibers alone is 257 ± 63 nm, chitosan’s inclusion in the solution led to a decrease in diameter due to raising the concentration of surface charges on the jet because of its cationic polyelectrolytic properties in acidic environments. Nanofibers of 30/70 chitosan/PVA ratio were found as an optimum product with a more uniform fiber distribution with a 223 ± 50 nm average fiber diameter but lower tensile strength than pristine PVA samples. Sajeev et al. also developed PVA–chitosan nanofibers for medical purposes and discussed the effect of the voltage, tip-to-target distance, and flow rate on fiber morphology [95]. It has been shown that the addition of chitosan to the structure reduces the fiber diameter and the lowest fiber diameter is obtained at a ratio of 8:2 PVA:chitosan formed with 8% PVA at a distance of 8 cm under 15 kV voltage. Raska et al. blended silk fibroin with PVA and produced nanofibrous scaffolds to replicate the radial, circumferential, and random directions of collagen fiber in the meniscus [148]. The scaffolds were examined only in terms of mechanical and morphological properties. Eggshell-derived calcium phosphate (TCP) and carbon dots, known for their applications in intra-cellular bioimaging and biosensors, were incorporated into poly(ε-caprolactone) (PCL)/PVA nanofibrous scaffolds to impart osteoinductive properties [196]. The addition of 1 wt.% carbon dots and specific TCP individually to PCL/PVA nanocomposite enhanced bone growth (ALS) activity and cell proliferation. The synergistic effect of carbon dots and TCP resulted in the highest osteogenic differentiation and proliferation rates compared to those of other scaffolds. Liu et al. investigated the feasibility of assembling small diameter tubes with helically arranged fibers to mimic the helical structure of native blood vessels [121]. The study used photo-crosslinkable PVA with styrylpyridinium pendent groups (SbQ) to produce nanofibers. Endothelial cells demonstrated a preference for attaching to PVA-SbQ fibers. The experiment revealed that endothelial cells were guided to grow along the PVA-SbQ fibers, emphasizing the potential and advantage of using these fibers in vascular grafts. Ngadiman et al. pointed out the difficulty of producing three-dimensional (3D) tissue engineering scaffolds, especially for hard tissues and proposed a technique that combines fused deposition modeling 3D printing with electrospinning [195]. Electrospun PVA/maghemite fibers are layered onto a 3D printed structure, forming a tissue engineering scaffold with milli- and microporous internal structures and a nanoporous external structure. Analyses revealed properties suitable for hard tissue engineering, including a compressive strength of 78.7 ± 0.6 MPa, a Young’s modulus of 1.43 ± 0.82 GPa, and biocompatibility to human fibroblast cells.

### 4.5. Wound Dressing

Wound dressings can be produced using electrospinning and electrospraying methods, which might be included under the classification of textile-based modern wound dressings. They have become increasingly popular, and there has been an increase in scientific studies on these products in recent years. Due to the wide range of uses, lack of a hazardous solvent, and good biocompatibility, studies on the electrospinning of PVA have primarily focused on biomedical applications. Xu et al. made the in situ application of electrospun PVA dressings with a hand-held device on the wound possible [197]. The device is operated with the push of a finger, and distance information and feed rate are not provided. The wound dressing was electrospun by 8 wt.% high molecular weight PVA in phosphate-buffered saline (PBS), including bone marrow stem cells with a 133.4 ± 29.6 µm fiber diameter. In vitro tests on cell growth and cytotoxicity demonstrated the dressing’s high biocompatibility; the cell survival rate was more than 90% and tended to increase in the following days. It was stated that using a relatively low voltage (10 kV) for fiber drawing did not adversely affect cell viability. According to in vivo tests, wound repair and collagen deposition were significantly faster in cell-included samples than in PVA alone and much faster in control samples. Chen et al. also utilized stem cell therapy with electrospun PVA mats. Cell electrospinning provided the advantages of achieving high cellular density, infiltration, and uniform distribution, facilitating functional connections between cells [198]. PVA mats served as an effective carrier for encapsulating adipose-derived stem cells, promoting their homogeneous distribution within the nanofibrous membrane. Encapsulating cells in PVA during in vitro cultivation enabled adherent growth on the membrane post PVA degradation, continuing for a certain duration, with cells still enveloped by PVA after 24 h. Remarkably, their vitality showed a significant improvement, reaching 133% after a 28-day culture period.

Moradzadegan et al. successfully electrospun PVA nanofibers containing acetylcholinesterase (AChE) and bovine serum albumin (BSA) as an enzyme-stabilizing component [97]. It was reported that the resulting fibers were approximately 190 nm in diameter, and by adding BSA and AChE, spread and more irregular fibers were obtained. In another study, Fatahian et al. included tranexamic acid, a blood coagulation active agent, of 10 and 20 mg/mL, and ceftriaxone, an antibacterial agent, of 0.1, 1, and 8 µg/mL into PVA nanofibers [98]. The average fiber diameter of the PVA nanofibers from 5% PVA was 250 ± 84 nm. The ceftriaxone, presented in powdered form, resulted in a viscosity rise, leading to an increase in the nanofiber diameter of up to 379 ± 84 nm. On the other hand, the tranexamic acid addition caused a decrease in viscosity, which may have disrupted the chain involvement of polymer chains that resulted in a lower fiber diameter (up to 110 ± 44 nm). The final products showed acceptable antibacterial and anti-coagulative properties. 

Koosha and Mirzadeh proposed PVA/CS nanofibers for biomedical applications [99]. The ideal product promoted L-929 fibroblast cell adhesion and growth without cytotoxic effects. Kang et al. heat-treated PVA nanofibers to limit solubility and coated them with chitosan, and demonstrated accelerated in vivo wound healing [199]. El-Attar et al. examined wound healing and antibacterial activity of the silver/snail mucous PVA nanofibers [200]. With snail mucus and silver nanoparticles introduced into the structure, the mean fiber diameter decreased from 170 nm to 126 nm and to 110 nm, respectively. In the first six hours, there was a rapid release of silver nanoparticles of up to 80% and a continuous release in the following 78 h.

Arun Karthick et al. were able to blend PVA with collagen using laboratory-extracted fish scale and commercial chitosan for electrospinning [201]. Optimum round, electrospun PVA fibers were obtained with 10 wt.% PVA with an average diameter of 142 nm. It is possible to add collagen in the 9:1 and 8:2 PVA:collagen ratios. This resulted in more flattened fibers but with lower diameters of up to 56 nm. It was also possible to electrospin 9:1 and 8:2 PVA: chitosan ratios for the chitosan blends. The average diameter of the chitosan-blended PVA nanofibers was less than the collagen-blended ones (47 nm). Both collagen- and chitosan-blended electrospun surfaces showed higher tensile strengths than the pristine PVA matrices because they had weaker moisture-absorbing properties. In addition, blending PVA with collagen and chitosan introduced antibacterial properties to nanofibrous surfaces. Zou et al. prepared PVA/chitosan (PVA/CS) nanofibers with antimicrobial effects, effectively promoting skin wound healing [202]. Gilotra et al. underlined the importance of PVA in advancing silk sericin-based nanofibrous mats for chronic wounds [203]. PVA facilitated the electrospinning of sericin in an 8:1 (*w*/*w*) ratio by optimizing solution properties for electrospinning. Employing PVA in silk sericin-based nanofibrous mats played a pivotal role in enhancing their water absorbency and serving as a substrate for the effective release of sericin. Sequeira et al. produced a nanofibrous membrane for dual purposes: skin regeneration and antimicrobial action [204]. The membrane was composed of PVA and Lysine, a protein that plays a key role in the formation of collagen. Additionally, it incorporated both anti-inflammatory agents (ibuprofen doped in the polymer solution) and antibacterial agents (lavender oil treatment). Polyethylene oxide solution was added to the polymer solution to reduce viscosity, facilitating the electrospinnability by adjusting the low viscosity, electrical conductivity, and decreasing surface tension. The incorporation of ibuprofen resulted in a decrease in all physico-chemical values. Despite the addition of ibuprofen, beadless nanofibers were produced, leading to a decrease in fiber diameter and a minor reduction in mechanical properties. Lavender oil-treated nanofibers, responsible for the main antibacterial action, exhibited a more irregular and rougher surface and a significant decrease in mechanical properties, particularly in terms of Young’s modulus and tensile strength. The electrospun mats of PVA-Lysine showed excellent biocompatibility and proved to be suitable for drug delivery, promoting their potential effectiveness as wound dressings. Mouro et al. incorporated *Chelidonium majus* L. extract, a medicinal plant, into polycaprolactone/PVA/Pectin nanofibrous mats using needleless emulsion electrospinning [205]. Following the incorporation of the extract, the nanofibers exhibited insignificant decreases in mechanical behavior, along with antibacterial activity against both *S. aureus* and *P. aeruginosa*. No cytotoxic effects were observed over a 7-day period. Moreover, there was a sustained release (of ~66%) of *C. majus* extract throughout the entire 30-day testing period.

### 4.6. Drug Delivery

In this subsection, we explore the diverse scope of drug delivery strategies using PVA electrospinning. Our review covers technologies, formulations, and approaches that optimize the administration and absorption of therapeutic agents, ranging from nanocarriers to targeted delivery systems, within the broader spectrum of electrospun PVA drug delivery.

The flexibility of electrospinning technology enables the creation of composite nanofibrous mats with improved properties for drug delivery. Electrospun PVA mats have been utilized as drug delivery reservoirs for the controlled and localized release of drugs, ensuring sustained and targeted delivery to specific sites, which is essential for enhancing therapeutic outcomes [206,207,208]. Ghalei et al. developed an innovative nanofibrous material by incorporating diclofenac-loaded zein nanoparticles into PVA. This novel dressing was designed to release diclofenac and offer an anti-inflammatory effect at the wound site. Incorporating zein nanoparticles into PVA nanofibers significantly reduced Young’s modulus. Release studies indicated that the embedded nanocontainer efficiently minimized the burst effect of diclofenac from zein nanoparticles, prolonging the release period to 5 days. In cell culture experiments, the dressing proved biocompatible, and the addition of nanoparticles to PVA nanofibers enhanced cellular proliferation and spreading. The hierarchical nanostructure synergistically combined the beneficial properties of zein NPs and PVA nanofibers, enhancing the dressing’s functionality by transferring the advantageous attributes of the encapsulated drug from the nanoparticles to the fibers [209]. Jannesari et al. blended PVA and poly(vinyl acetate) and demonstrated a useful and convenient method for electrospinning in order to control the rate and period of drug release in wound healing applications [207]. Yang fabricated nanofibrous surfaces comprising chitosan, PVA, and graphene oxide (GO), loaded with antibiotic drugs, including Ciprofloxacin (Cip) and Ciprofloxacin hydrochloride (CipHCl) [210]. The effective loading of antibiotic drugs into the nanofibers, with a portion absorbed into GO nanosheets, was achieved. Interestingly, the release of drugs absorbed in the GO nanosheets regulated the profile, preventing the initial burst release, and the addition of GO slightly improved the release ratio. The nanofibrous membranes exhibited significantly enhanced antibacterial activity against *Escherichia coli*, *Staphylococcus aureus*, and *Bacillus subtilis* with excellent cytocompatibility for Melanoma cells.

Cui et al. used PVA/chitosan nanofibers for transdermal drug delivery [211]. A glutaraldehyde-crosslinked electrospun surface presented an ideal surface for ampicillin sodium with a lower drug release rate and smaller burst release than uncrosslinked samples. Yadav et al. investigated the potential of CS/PVA-blended fibers loaded with curcumin (CUR) and zinc oxide (ZnO) for healing diabetic foot ulcers [212]. Crosslinked CUR-ZnO-CS/PVA fiber mats exhibited controlled drug release for 72 h, and were confirmed as non-toxic against HaCat cells via cytotoxicity analysis. The optimized nanofiber membrane displayed antimicrobial activity against *S. aureus* and *P. aeruginosa*. In vivo wound healing analyses demonstrated superior anti-inflammatory effects and an enhanced wound contraction ability, indicating its treatment potential. Rahmani et al. investigated the pH-responsive release of CUR from the PVA-graphene oxide (GO)-silver (Ag) nanofibers containing CUR nanofiber mats [213]. They observed a fine porous morphology with potent antibacterial activity against *E. coli* and *S. aureus*. The incorporation of Ag nanoparticles into the nanofibers increased loading and encapsulation efficiencies. The pH sensitivity of PVA/GO-Ag-CUR nanofibers was confirmed, showing an inhibited CUR release at pH 7.4 and enhanced release at pH 5.4. In vitro wound healing tests on NIH 3T3 fibroblast cells demonstrated accelerated growth and proliferation on PVA/GO-Ag-CUR nanofibers.

Hussein et al. investigated a wound healing drug delivery system utilizing *Punica granatum* L. extract [214]. The extract, combined with chitosan–gold nanoparticles, was loaded into PVA fibers (at 0.1–0.9% *w*/*v*). The glutaraldehyde crosslinking of PVA fibers enhanced mechanical strength, preserved porosity, and improved drug release. The drug release pattern followed a Fickian diffusion mechanism. The formulation, incorporating 0.9% nanoparticles, exhibited long-term stability, the highest antibacterial activity, excellent biocompatibility, and robust cell adhesion and proliferation.

### 4.7. Cancer Therapy

Focusing specifically on cancer therapy, this section provides a comprehensive review of the current state of therapeutic interventions for cancer using PVA electrospinning. We discuss the latest breakthroughs in targeted therapies, immunotherapies, and precision medicine tailored to combat the complexities of cancer.

Electrospun PVA mats are drawing interest in other biomedical research, particularly in cancer therapy. These nanofibers serve as effective substrates for cancer drugs, showing potential for targeted drug delivery to tumors. Chee et al. incorporated doxorubicin hydrochloride (DOX), a chemotherapy drug, into PVA mats, and annealed them thermally with a polyacrylic acid electrospun layer that carried clarithromycin, an antibiotic. When using DOX, pure PVA demonstrated a notably slow release rate (~2%) in 24 h, which was further reduced with annealing (~1%) [215]. This suggests the controlled and non-toxic release of DOX, which is important for targeted applications. PVA mats with DOX demonstrated promising inhibition of osteosarcoma cancer cells, achieving up to 60% inhibition. Cao et al. employed DOX in core-shell PVA/silk fibroin nanoparticles produced through electrospraying [216]. The drug reached over a 90% encapsulation efficiency, and controllable release profiles were achieved by modifying the polymer ratio, minimizing the initial burst release through fibroin coating. Sustained DOX release yielded an apoptosis rate of up to 35% in tumor cells.

Electrospinning assisted in producing gold nanorods embedded in PVA/CS hybrid nanofibers. The gold nanorods and DOX-infused fibers effectively inhibit ovarian cancer cell growth and proliferation. Additionally, their unique optical properties make them suitable as cell imaging agents [217]. In another study, 2D molybdenum disulfide (MoS2), a photothermal transforming agent, and DOX were co-loaded into PVA/CS nanofibers to inhibit postoperative tumor reoccurrence [218]. These nanofibers exhibited an exceptional photothermal conversion capability, achieving a conversion efficiency of 23.2% and complete inhibition of postoperative tumor reoccurrence. Additionally, PVA was introduced into the core solution to facilitate the co-axial electrospinning of 5-Fluorouracil with PCL, enhancing the electrospinnability compared to the tendency of the drug solution alone to spray. Incorporating PVA in the core led to a lower encapsulation efficiency (52.71%) with nanocrystals outside the sheath. Removing PVA in the core and using a lower voltage (1.4 kV/cm) improved drug encapsulation to 77.5% [219].

## 5. Conclusions

This article focuses on using PVA to develop electrospun structures with potential applications in the biomedical field. PVA is a versatile polymer used as a base for various electrified structures. Recent papers have highlighted the process modifications required to achieve controlled morphologies and the encapsulation of active ingredients. The control of morphologies depends on process parameters such as flow rate, applied voltage, needle size, and distance between the needle and collector and formulation parameters such as the molecular weight of the polymer and drug, solution viscosity/volatility, surface tension, and viscosity while taking into account the problem of the degree of hydrolysis of PVA, which plays a significant role in the physico-chemical properties of solutions.

This polymer has been widely used for filtration, gas sensors, wound dressings, tissue engineering and scaffolding, drug delivery, and, more marginally, cancer therapy, thanks to its hydrophilicity, biocompatibility, non-toxicity, and mechanical strength. Its mechanical and structural properties can also be tailored to specific applications by blending it with other polymers, such as chitosan, or introducing nano-objects.

In recent years, the versatility of the electrospinning process has enabled the modification of resulting structures’ morphology to align with desired properties, such as promoting cell proliferation on surfaces or controlling the release of active substances. The development of new processes associated with new morphologies or new surface states represents the most innovative research opportunities. Nevertheless, the development of new processes based on electrospinning requires systematic studies on the process and formulation parameters, including rheological studies on polymers in solution for polymer blends or polymer-active ingredient blends to control interactions between compounds during the process.

Anticipated advancements in electrospinning technology point toward several promising avenues for enhancing the functional properties of PVA-based electrospun structures. The exploration of co-axial or tri-axial systems holds potential for creating innovative morphologies, while an alternative path involves the formation of microfibers at a scale of a few micrometers. This microfiber approach, once entangled, could pave the way for the development of smart coatings with an expanded specific contact surface.

Moreover, a distinct strategy focuses on scaling up electrospinning processes to industrial levels, achieved by enhancing the production capacity through the application of diverse spinneret designs. This industrial-scale production not only addresses quantity but also opens doors for efficient and large-scale manufacturing.

Looking ahead, the evolution of electrospinning technology may usher in a new era of personalized wound care. Tailoring electrospun structures to meet individual patient needs could become a reality, presenting an exciting frontier in the development of advanced and patient-specific biomedical products.

## Figures and Tables

**Figure 1 ijms-25-01668-f001:**
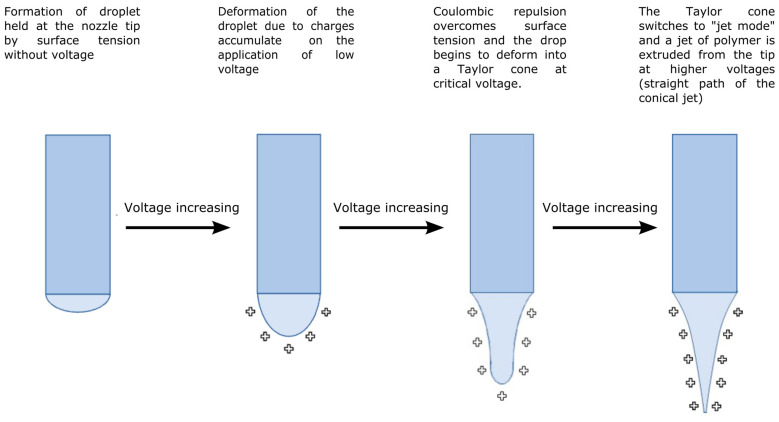
Deformation modes of the polymer droplets or jet during the electrospinning process modified after [76,77].

**Figure 2 ijms-25-01668-f002:**
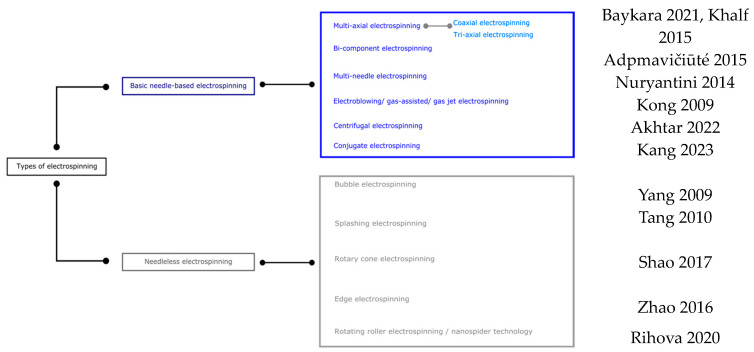
Classification of electrospinning methods for PVA [80,81,82,83,84,85,86,87,88,89,90,91].

**Figure 3 ijms-25-01668-f003:**
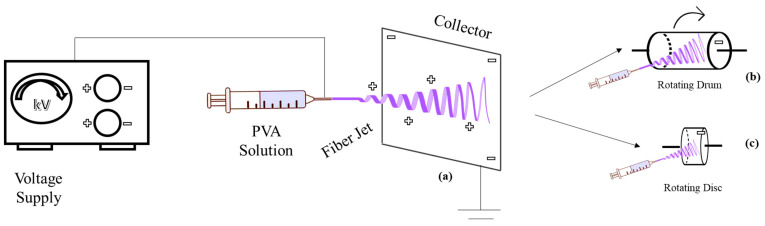
Horizontal electrospinning setup with different collectors: (**a**) static collector, (**b**) rotating drum collector, (**c**) rotating disc collector.

**Figure 4 ijms-25-01668-f004:**
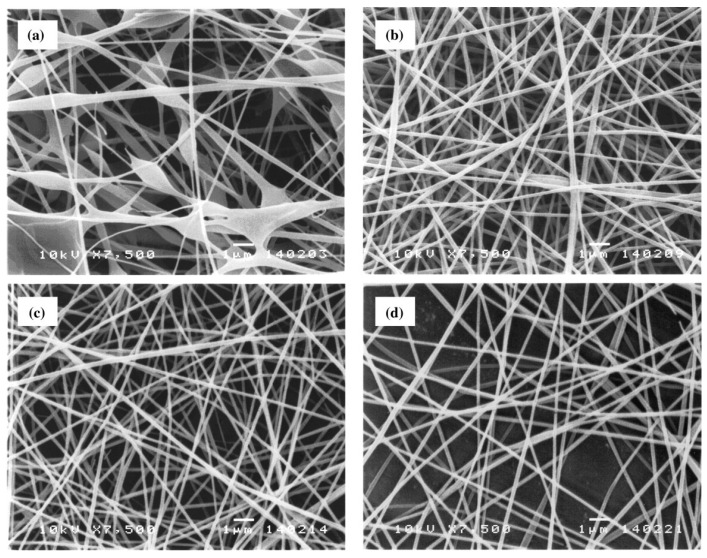
Selected SEM images (magnification = ×7500 and scale bar = 1 µm) of the as-spun fibers from sonicated 10% *w*/*v* PVA solution. An electrical potential of 15 kV was applied over varying collection distances of (**a**) 5, (**b**) 10, (**c**) 15, or (**d**) 20 cm, and the feed flow rate was 1 mL h^−1^ (reproduced from reference [13] with permission).

**Figure 5 ijms-25-01668-f005:**
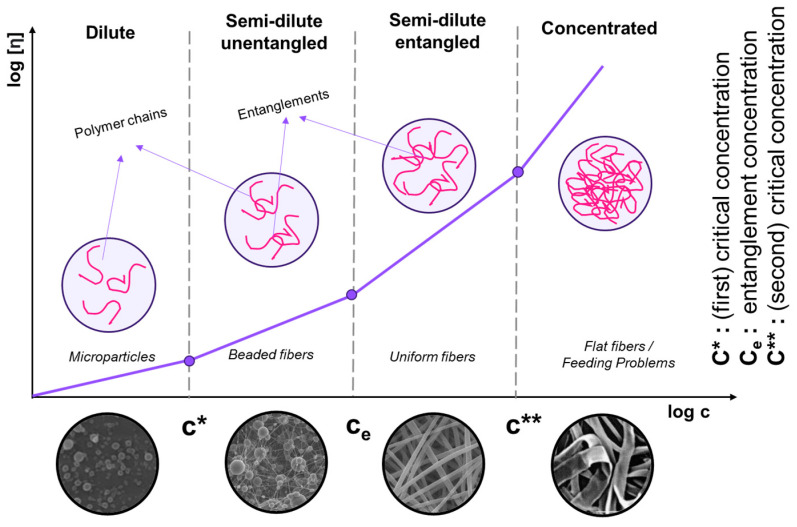
Concentration dependence of the polymer solution. Scaling relationship between the critical concentration (C*), chain entanglement concentration (Ce), and viscosity for various concentration regimes.

**Figure 6 ijms-25-01668-f006:**
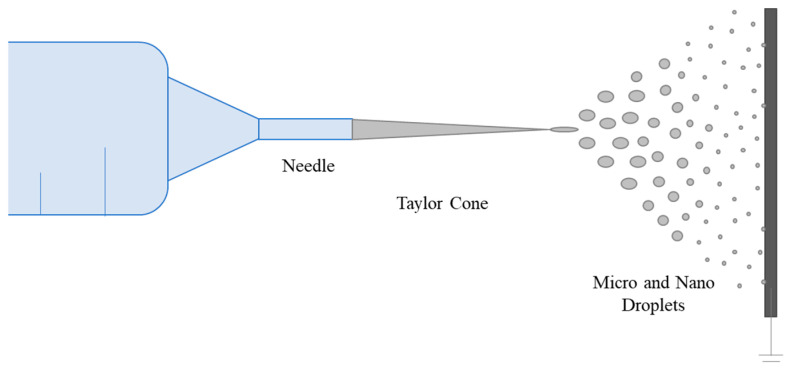
Schematic representation of the electrospraying process.

**Figure 7 ijms-25-01668-f007:**
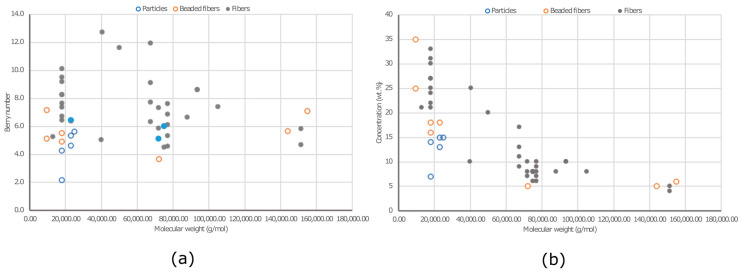
Morphologies obtained using different concentrations of PVA on different Mws: (**a**) Berry number vs. Mw and (**b**) concentration (wt.%) vs M_W_ (g/mol).

**Figure 8 ijms-25-01668-f008:**
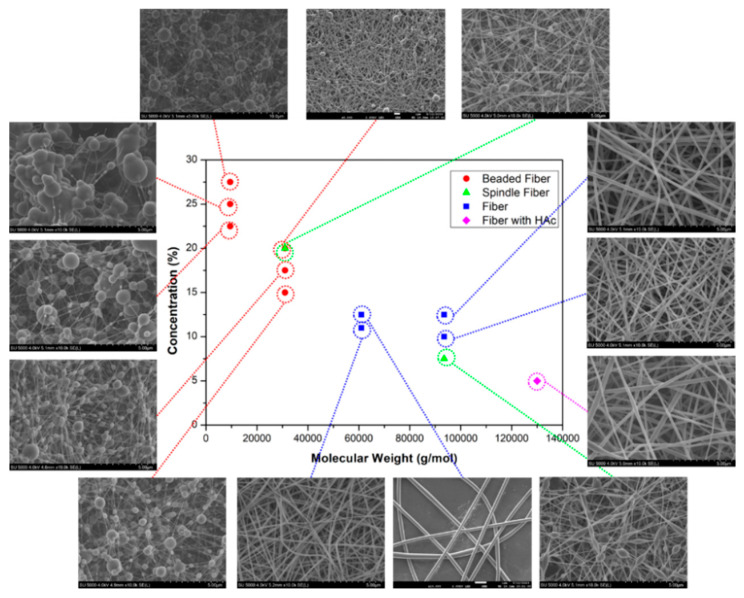
SEM micrographs of electrospun PVA mats with different concentrations (wt.%) and M_W_s (g/mol) [161].

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
