# Peer review of "PVA-Based Electrospun Materials—A Promising Route to Designing Nanofiber Mats with Desired Morphological Shape—A Review"

_ijms, 2024, doi:10.3390/ijms25031668_

Round 1
Reviewer 1 Report
Comments and Suggestions for Authors
The submitted manuscript consists of two interlacing topics: presentation of the process of electrospinning and its attributes, and an attention to the applications based on PVA usage. Unfortunately the manuscript exhibits a series of basic shortcomings|:
- These two inputs are not well balanced. The basic principles of electrospinning should be based on the basic literature sources and not on the repeatedly re-written paragraphs in the papers devoted to the specific problems but not contributing to the electrospinning roots.
- A review paper should not introduce only a cumulation of the published papers using PVA but also to redistribute them and to point out the most beneficial ones. Otherwise it does not make it easy for a reader to be oriented in the topic and a list of such papers he/she can find himself/herself immediately in various literature sources such as WoS.
- Not all crucial parameters are mentioned in the Introduction as e.g. temperature and humidity (l. 114).
- In the text an emphasis is often paid to an interplay of two chosen parameters. However, a situation is much more complicated and usually there is involved more than two attributes. Of course, it is not possible to take into account simultaneously more parameters but there are studies combining more than only two parameters. It should be also projected into the manuscript.
- Some materials cannot be electrospun. However, after adding a relatively small amount of excellently electrospinnable materials such as PVA, PVB, PEO the problem can be very successfully solved with really small influence on the ´unsuitable material´ characterization. Such paragraph could be also inserted.
- Figure 2: It should be beneficial to complete the individual lines with the representative references.
- Figure 3: An initial straight segment of the jet is completely missing; the last part is swirling not waving.
- l. 153: An optimal situation is when a solvent is evaporated during a jet path to the collector.
- ll. 183-192: The impacts of the individual forces differ. It should be beneficial to introduce just briefly their comparison or significance (e.g. the gravitational force with the other ones).
- l. 224 and elsewhere: just to present two numbers (without other context) does not provide sufficient information. A drop or increase in percentage can (maybe) give a better insight.
- l. 269: Can it result in electrospraying instead of electrospinning after solvent evaporation?
Based on the above comments I cannot recommend the manuscript in its present form for publication in the journal IJMS.
Author Response
Dear Reviewer,
First of all, we would like to thank you for your constructive and helpful comments on our manuscript entitled " PVA-based electrospun materials - A promising route to designing nanofiber mats with desired morphological shape - A review". We believe that by carefully addressing all most of your comments, we have indeed made important improvements to the manuscript.
Attached to this letter and to our revised manuscript are our detailed responses to each comments (our corresponding changes to the manuscript are marked in blue in the revision and are referenced here). We hope that you will find in our responses the clarifications and improvements you have requested, and we look forward to their assessment of our revised submission and await your considered decision.

Reviewer 2 Report
Comments and Suggestions for Authors
The review article discusses recent advancements in poly(vinyl alcohol) electrospinning, covering process parameters, solution characteristics, and environmental conditions affecting the properties of resulting matrices. Emphasis is placed on biomedical applications, including wound dressings, drug delivery, tissue engineering, and biosensors. The study also introduces the formation of particles via electrospraying of poly(vinyl alcohol). It provides a critical evaluation of the pros and cons of different poly(vinyl alcohol) matrices, offering recommendations for future research in this domain.
The article deals with a topic of a wide interest for the readers. Please find below my comments and suggestions:
Lines 60-63. Please provide an example regarding the two-digits used for the PLA notation. The sentence used in lines 61-63 is the same used in a white paper from Aldrich-Merck entitled: "Polyvinyl alcohol: Revival of a long lost polymer" by Adela Kasselkus, Erica Weiskircher-Hildebrandt, Eva Schornick, Finn Bauer, Mengyao Zhen. Although it is a well-known nomenclature, it should be explained in the paper, given that it is a review.
Sometimes it is worth providing a list of acronyms in the document so that the terms are precisely defined.
Even though well-known acronyms are used, they should be defined the first time they are used, see for example TCD (Lines 239-242).
In line 249 – 250, it says: “In Phachamud's experiment, the difference value of distance was relatively small, and the apparent effect on fiber characteristics was not found [81]”. Why was it not found? Is there any explanation?
In lines 252-253, it is written “On the one hand, it has been observed that the diameter of PVA nanofibers increases with increasing tip-collector distance [86]” but then it is mentioned that Suphaphol explains that there is a decrease in in fiber diameter with increasing distance. The sentence referred to reference 86, is correct? Does it increase or decrease?
Regarding Figure 4, the scale bar is not clearly visible. If possible, please ask for permission to modify the image and published so that the scale bar is properly visualized.
Please kindly provide references to support the sentence in lines 339 - 340. Mw, write the full word and define the term.
Lines 392-393, please define MW, DH, concentration and write the terms properly. Referred to concentrations, please clarify the units: a) in line 62, it is mentioned 4% solution (with no units, %, wt/wt, % wt/mL, % v/v?); b) in line 398, 0.1 to 0.2 wt% (is it wt/wt?); in line 408, it is written 1% without units... Please clarify.
Kindly revise the rest of the document to clarify and unify nomenclature regarding the use of acronyms and their proper definition.
Author Response

(The authors gave the same response as above.)

Reviewer 3 Report
Comments and Suggestions for Authors
1. Certain references should be cited to support the statements, which include Line 99 – 113, Line 159 – 168, Line 226 – 230, Line 277 – 284, Line 325 - 337
2. Figure 2 couldn’t provide enough useful information. It is suggested to cite certain references to support each system configuration.
3. Permission for resuing the figures should be obtained, including Figure 1, Figure 3, Figure 5, Figure 6, Figure 7, and Figure 8.
4. Surface tension should not be listed as an independent parameter, since it is affected by polymer concentration and solvent. Consider incorporating this information into other sections.
5. Certain figures about the applications should be used for better understanding.
6. Tissue engineering is broad and general. Wound dressing can also be taken as one part of tissue engineering. The authors should make clarification on this.
7. The examples introduced in cancer therapy are also suitable for drug delivery. The authors should make clarification on this.
Comments on the Quality of English Language1. The authors failed to properly introduce the abbreviations used in this article. In the abstract, PVA should be introduced. The full name of chitosan, Mw, and RH should be provided when they are introduced for the first time.
2. Use solely electrospin instead of electro-spin, and use solely electrospraying instead of electro-spraying.
3. Line 115 should be a new paragraph.
4. Line 107 : ‘usually adjusted to’ → ‘usually are adjusted to’
5. Line 233: The unit for the voltage should be provided.
6. Line 234: Delete ‘respectively’.
7. Line 462: ‘although’ → ‘though’
8. Line 521: ‘although’ → ‘though’
9. Line 530: ‘, marks the’ → ‘marks the’
10. Line 582: ‘PVA fiber morphology about’ → ‘The dependence of PVA fiber morphology on’
11. Line 618: ‘is fundamental application area’ → ‘is a fundamental application area’
12. Line 728: Avoid using two ‘which’ in one sentence.
Author Response

(The authors gave the same response as above.)

Round 2
Reviewer 1 Report
Comments and Suggestions for Authors
The authors met all the comments raised in my preceding review